# Fifty Years of Change in a Coniferous Forest in the Qilian Mountains, China—Advantages of High-Definition Remote Sensing

**Shu Fang [1] and Zhibin He [2],***

1 College of Urban, Rural Planning and Architectural Engineering, Shangluo University, Shangluo 726000, China; fangs@lzb.ac.cn
2 Linze Inland River Basin Research Station, Chinese Ecosystem Research Network, Key Laboratory of Eco-hydrology of Inland River Basin, Northwest Institute of Eco-Environment and Resources, Chinese Academy of Sciences, Lanzhou 730000, China
* Correspondence: hzbmail@lzb.ac.cn; Tel.: +86-136-6930-4220

**Abstract:** Mountain ecosystems are significantly affected by climate change. However, due to slow vegetation growth in mountain ecosystems, climate-induced vegetation shifts are difficult to detect with low-definition remote sensing images. We used high-definition remote sensing data to identify responses to climate change in a typical *Picea crassifolia* Kom. forest in the Qilian Mountains, China, from 1968 to 2017. We found that: (1) *Picea crassifolia* Kom. forests were distributed in small patches or strips on shaded and partly shaded slopes at altitudes of 2700–3250 m, (2) the number, area, and concentration of forest patches have been increasing from 1968 to 2017 in relatively flat and partly sunny areas, but the rate of area increase and ascend of the tree line slowed after 2008, and (3) the establishment of plantation forests may be one of the reasons for the changes. The scale of detected change in *Picea crassifolia* Kom.forest was about or slightly below 30 m, indicating that monitoring with high-resolution remote sensing data will improve detectability and accuracy.

**Keywords:** mountain coniferous forest; *Picea crassifolia* Kom.; high-resolution remote sensing

---

## 1. Introduction

The temperature increase associated with global climate change was three times higher in mountain areas in the past 40 years than the global average, and the frequency of extreme climate events was also significantly greater in mountainous than in other ecosystems [1]. The tree line is the continuous forest boundary of a mountain forest, and it is the transition zone from forest to tundra [2–4]. Mountain vegetation, especially at the alpine tree line, is highly sensitive to climate change and can reflect climate change more quickly than other types of vegetation [5]. In fact, the upper limit of the tree line is expected to shift up by as much as 300 to 600 m in elevation as a result of climate warming [6]. Further, deciduous forests may increase in presence at the alpine tree line due to warming, increase in length of the growing season, and decreased severity of the winter environment [7].

However, the upward movement of the alpine tree line caused by a temperature rise is relatively slow [8]; additionally, warming has been linked to drought stress at the alpine tree line, which may decrease the rate of forest growth [9]. The impact of the warming climate on coniferous forests was not significant because snow, wind damage, and frequency of pests and diseases were delayed in alpine areas of the Northern Hemisphere until the 1980s [10].

The Qilian Mountains are located in the arid northwestern region of China and play an important role in contributing water to the lowlands. Forest vegetation in the Qilian Mountains is a valuable forest resource with the ecological function of water conservation [11].

*Picea crassifolia* Kom. (Qinghai spruce) is the dominant tree species in the Qilian Mountains, forming an almost pure spruce forest in places [12]. *Picea crassifolia* Kom. has been widely used in dendroclimatology to determine the relationships between tree growth in the Qilian Mountains and climatic factors [13–15]. A recent survey showed that the *Picea crassifolia* Kom. tree line trended upward [16], but the sample size in the survey was limited. In efforts to determine the responses of Qinghai spruce forest to climate change, traditional dendroclimatology approaches result in great uncertainty and inconsistency with sample surveys. At the same time, research scale is not representative, remaining on a single tree. Thus, a larger-scale approach is needed to determine the response of *Picea crassifolia* Kom. forests to climate warming.

In mountainous areas where data are scarce and sites may be inaccessible, remote sensing data and GIS (Geographic Information System) technology are particularly useful in monitoring changes in forest location and landscape patterns [17,18]. Continuity in remote sensing data allows for dynamic monitoring of the tree line position and pattern change [19,20]. As early as 1995, tree lines of patch forest ecotones in the Rocky Mountain National Park were detected with aerial photographs of 1988 and 1990 [21]. The tree line can be directly detected based on the vegetation type over the remote sensing image, and the identification of the tree line can be more accurate with the use of the Normalized Difference Vegetation Index (NDVI) or topographic data such as elevation and slope [22,23]. Direct classification for Landsat images can be used for extracting tree line ecotone, but a complex method, such as linear spectral mixture analysis, should be used for detecting the alpine tree lines [24,25].

Changes in the tree line using remote sensing data are mainly identified with vegetation coverage or NDVI [26]. However, coniferous forests changed more slowly than tundra and shrubland in alpine tree line areas [26], and slow vegetation changes may be difficult to detect with low-resolution remote sensing data. Tree lines in coniferous forests in the Finnish Lapland have increased in tree density, and expanded upward in the last three decades, but that could not be detected with NDVI or land-cover classification from Landsat [27]. High-resolution remote sensing data make it possible to monitor some areas where the tree line does not change significantly even at species level, and where accuracy and precision of monitoring results will be improved [28]. The growth rate of *Picea crassifolia* Kom. natural forests in the Qilian Mountains is slow, and the natural renewal capacity of secondary forests is low [29], making it difficult to monitor forest changes using general remote sensing images, such as Landsat with a resolution of 30 m.

To improve monitoring of changes in *Picea crassifolia* Kom. forest, we chose high-resolution aerial photography and remote sensing images of the Dayekou watershed in the Qilian Mountains at a resolution of about 2 m for 1968, 2008, and 2017. We aimed to determine whether (1) the *Picea crassifolia* Kom. tree line in the Qilian Mountains changed in the past 50 years, (2) the landscape pattern of the tree line position has changed, and (3) changes were consistent with climate change. The results of this study provide a reference for high-resolution remote sensing image monitoring of the position and landscape pattern of forest tree lines for slowly growing coniferous forests in alpine mountain areas.

## 2. Materials and Methods

### 2.1. Study Area

We conducted this study at the Dayekou watershed (38°26′–38°35′ N, 100°14′–100°19′ E) in the upper reaches of the Heihe River Basin in the Qilian Mountains of China (Figure 1). It is located in the Xishui Forest District of the middle Qilian Mountains, and has an area of about 73 km$^2$ at an altitude of 2500–4700 m. Slopes of the low-mountain areas in the Dayekou watershed are between 10 and 30°, and those of the high-mountain areas are about 40°. Climate in the Dayekou watershed is dry and semi-humid, with average annual temperature of 2.0–3.5 °C, and average July temperature of 10–14 °C. Annual precipitation is 350–450 mm, which is concentrated in June–September, annual evaporation is 1050–1100 mm, annual cumulative duration of sunshine is 1890–1900 h, and average annual relative

humidity is 60% to 65%. The foundation species is *Picea crassifolia* Kom., which has patchy distribution on shaded and semi-shaded slopes at an altitude of 2500–3400 m. The areas also contain sparse Qilian juniper (*Sabina przewalskii* Kom.). The soil is mainly grey cinnamon and chestnut, and is characterized as relatively thin, with mainly silt sand texture [30].

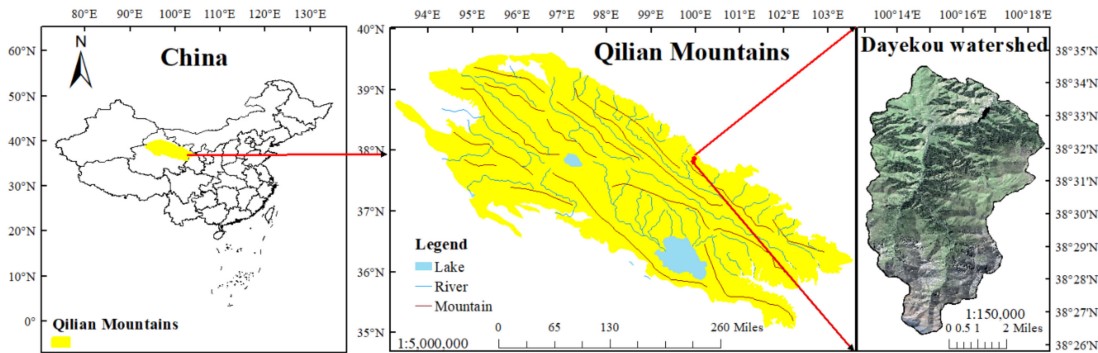

**Figure 1.** Location of Dayekou watershed in the Qilian Mountains of northwestern China.

### 2.2. Remote-Sensing Data Acquisition and Pre-Processing

We chose high-resolution remote sensing images of the Dayekou watershed for 1968, 2008, and 2017. Specifically, we used the 1968 keyhole historical aerial photograph data [31] with a resolution of 1.8 m, the 2008 Quickbird image data [32] with a resolution of 2.4 m, and the 2017 ZY-3 satellite images [33] with a resolution of 2.1 m. The images had the least cloud amounts and high data quality and were selected among satellite images of similar resolution for the same period.

Next, we selected data for accuracy verification and digital elevation model (DEM) data from the Heihe Data Management Center (http://www.heihedata.org/) (Figure 2). First, we used the following datasets to establish a database of field sampling points to verify the accuracy of remote sensing image interpretation: (1) a dataset of forest structure at the fixed sampling plot in the Pailugou watershed and Dayekou watershed foci experiment area from 2003 [34], (2) a dataset of forest structure at the fixed sampling plot in the Pailugou watershed and Dayekou watershed foci experiment area for 2007 [35], (3) a dataset of forest structure at the temporary forest sampling plot in the Dayekou watershed foci experimental area for 2008 [36], (4) the Heihe Integrated Remote Sensing Joint Test, Arid Region Hydrological Experimental Area and Forest Hydrological Experimental Area, Land Use and Land Cover Survey Dataset [37], and (5) a dataset of forest structure measurements for the fixed forest sampling plots in the Dayekou and Pailugou watershed foci experimental areas (2003–2007) [38]. Then, we used the 1 m digital elevation model (DEM) of the Dayekou watershed established by wordview-2 data in 2012 [39] as the basis for extracting relevant terrain factors.

To improve accuracy of remote sensing classification, we used drone image photographed by DJI (Da Jiang Innovations) phantom 4 pro for areas with plantations and Qilian Juniper forests (Figure 3).

After pre-processing of the three images for atmospheric interference, and georeference calibration, we used the k-means unsupervised classification method in ENVI 5.3 [40] to classify patchy vegetation features more accurately [41], and to divide the Dayekou watershed into coniferous and non-coniferous forests. Some Qilian juniper trees in the coniferous forest were eliminated by visual adjustment based on forest canopy closure, distribution, location, and data from drone surveys; then, the remaining coniferous forests were classified as *Picea crassifolia* Kom. The accuracy of the decoded image was verified based on field and drone sampling points of the Heihe Data Center. The kappa index was 0.8630, and total accuracy was 93.67%. Finally, the distribution maps of *Picea crassifolia* Kom. forests in the Dayekou watershed for the 1968, 2008, and 2017 were obtained, and the boundary of the forest is the tree line. We have counted the maximum, minimum, and average values of each forest patch to explore the upper, lower, and overall tree line changes.

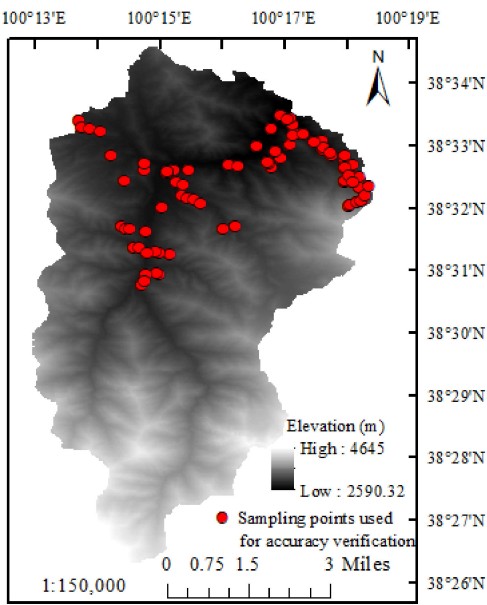

**Figure 2.** High-resolution digital elevation model (DEM) and verification points.

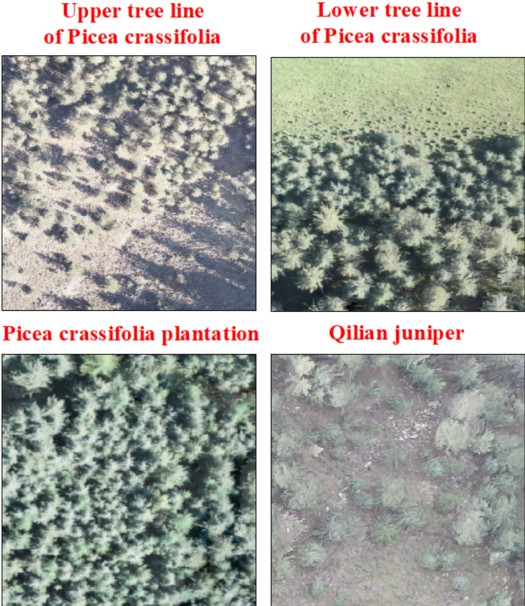

**Figure 3.** Verification data for drone sampling sites.

*2.3. Data Analysis*

2.3.1. Changes in the Distribution of *Picea crassifolia* Kom.

Regional statistics and spatial analysis for the three phases were performed with ESRI ArcMap 10.4 [42] raster data analysis to obtain characteristics of *Picea crassifolia* Kom. distribution. Topographic data were superimposed on the distribution data, and elevation, slope, and aspect characteristics of *Picea crassifolia* Kom. distribution were obtained for the three-phase remote sensing images.

The area conversion matrix for 1968, 2008, and 2017 remote sensing images of *Picea crassifolia* Kom. forest and non-*Picea crassifolia* Kom. forest was first calculated to describe the area change between Qinghai spruce forest and non-*Picea crassifolia* Kom. forest between time periods.

We used three terrain indicators, elevation, slope, and aspect, which were calculated by the 'Surface' tool in 'Spatial Analyst Tools' in ESRI ArcMap 10.4, to detect dynamic changes in forest

distribution. Then we used 'Reclassify' in 'Spatial Analyst Tools' to classify elevation, slope, and aspect. The forest was distributed at elevations between 2600 and 3550 m, so we divided elevation into 19 levels, with the first level at below 2600 m, one level added every 50 m, and the last level at >3450 m. Slope ranged from 0 to 83.99°, and we divided slope into eighteen 5° increments (categories). The aspect distribution of forests was mainly related to shaded and sunny slopes, so the slope direction was divided into 5 categories, with 0 for no slope direction, 1 for shaded slopes (N (0–22.5°, 337.5–360°)), 2 for semi-shaded (NE (22.5–67.5°), E (67.5–112.5°), NW (292.5–337.5°)), 3 for sunny (S (157.5–202.5°)), and 4 for partly-sunny slopes (SW (112.5–157.5°), SE (202.5–247.5°), and W (247.5–292.5°)).

Subsequently, we used the terrain distribution index P of elevation, slope, and aspect (Equation (1)) to analyze the relationship between distribution and topography of *Picea crassifolia* Kom. forest, and vegetation change rate index K (Equation (2)) to determine the rate of change of *Picea crassifolia* Kom. forest area between different periods.

$$P = S_{ie}/S_e, \tag{1}$$

$$K = (S_b - S_a)/S_a \times (1/t), \tag{2}$$

where: e represents topographic factors of elevation, slope, and aspect, $S_{ie}$ represents the area of *Picea crassifolia Kom.* forest at a specific level in e elevation, slope, and aspect, $S_e$ is the total area in a specific level of e elevation, slope, and aspect within the entire study area, $S_a$ and $S_b$ are the area of vegetation at the beginning and end of the study period, and t is the length of the study period.

### 2.3.2. Landscape Changes of *Picea crassifolia* Kom.

Landscape index was used to describe landscape patterns and their changes, and to establish the relationship between landscape pattern and process, as it is the most commonly used quantitative research method in landscape ecology [43]. To calculate the landscape pattern index, the scale of calculation is determined first. Based on the results of extensive landscape-scale research [44,45], and conditions of the selected research area, 20 scales were selected from 1 to 30 m, with a 1 m interval between 1 and 10 m, and a 2 m interval between 10 and 20 m. We calculated the landscape pattern index at different scales in FRAGSTSTS 4.2 [46] to obtain the response map at different landscape pattern scales. We selected an optimal scale for our study, and, at this scale, we calculated the landscape pattern index of *Picea crassifolia* Kom. and analyzed changes within it for the last 50 years. In this study, we calculated six landscape pattern indices (Table 1) for *Picea crassifolia* Kom. forest at the landscape level [47–50].

**Table 1.** Landscape pattern indices.

| Metrics | Name | Description |
|---|---|---|
| Area and Edge | Largest Patch Index (LPI) | The proportion of the largest patch area |
| Shape | Perimeter-Area Fractal Dimension (PAFRAC) | Non-randomness or degree of aggregation for different patches |
| Aggregation | Patch Density (PD) | Number of patches per unit area |
| | Splitting Index (SPLIT) | The number of patches in a landscape divided into equal sizes keeping landscape division constant, express the separation degree of individual distribution in different |
| | Aggregation Index (AI) | The degree of aggregation of similar patches |
| | Landscape Shape Index (LSI) | Continuity and complexity of landscape shape and the measurement of the perimeter-to-area ratio for the landscape as a whole |

## 3. Results

### 3.1. Shift in the Forest Cover

The *Picea crassifolia* Kom. forest in the Dayekou watershed was distributed in patches and strips (Figure 4). Most patches were distributed in the northeast-southwest direction and were found mostly in the north of the basin. Overall, the number and area of *Picea crassifolia* Kom. patches in the basin have been increasing since 1968: there were 222 forest patches in 1968, 263 in 2008, and 264 in 2017. Forest area in 1968 was 12.44 km$^2$, accounting for 17.04% of the entire basin area. By 2008, forest area increased to 14.94 km$^2$, and the proportion of total forest area has increased by 3.43%. By 2017, forest area reached 15.48 km$^2$, accounting for 21.21% of the total watershed area. The average area of forest patches in Dayekou remained at 0.056–0.058 km$^2$ for 50 years. Forest patches with areas smaller than the average accounted for more than 80% of the total number of patches.

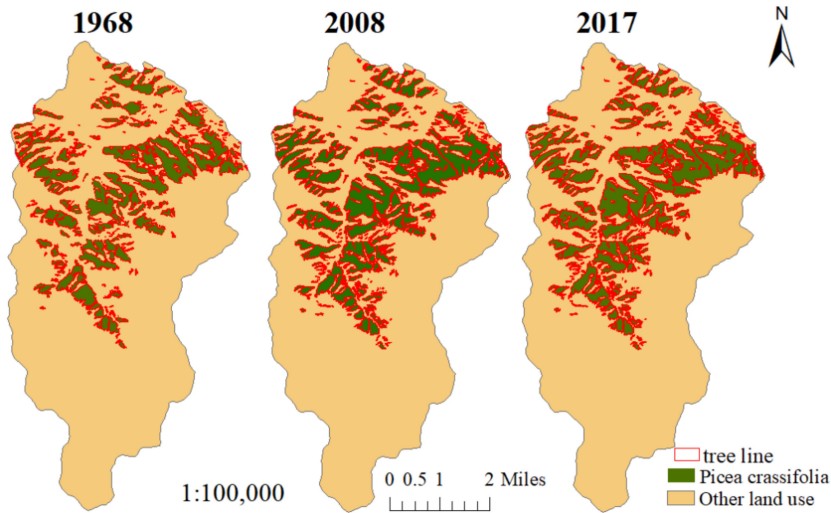

**Figure 4.** Distribution of *Picea crassifolia Kom.* in the Dayekou watershed in different years.

The spatial distribution change of *Picea crassifolia* Kom. forest area from 1968 to 2017 is shown in Figure 5. Between 1968 and 2008, patch areas that remained unchanged accounted for 92.14% of the total area of the watershed. During that time, the ratio of *Picea crassifolia* Kom. area to non-*Picea crassifolia* Kom. area was 2.24%, while conversion of areas from non-*Picea crassifolia* Kom. to *Picea crassifolia* Kom. reached 5.62%.

Between 2008 and 2017, the area where the *Picea crassifolia* Kom. forest remained unchanged accounted for 97.49% of the total area of the basin. During that time, the proportion of *Picea crassifolia* Kom. area converted to non-*Picea crassifolia* Kom. was 0.89%, while the proportion of non-*Picea crassifolia* Kom. converted to *Picea crassifolia* Kom. area was 1.62%. Further, the most *Picea crassifolia* Kom. forest changes in area occurred at the boundaries of forest patches.

The rate of increase in patch area of the *Picea crassifolia* Kom. forest was 50.30 m$^2$/per year from 1968 to 2008, and it decreased to 40.42 m$^2$/per year from 2008 to 2017.

The regulation of elevation and aspect of the distribution of *Picea crassifolia* Kom. forest is shown in Figure 6. The altitude range of forest distribution in 2017 was 2639.70 to 3412.80 m, with the mean elevation of the distribution of 266 forest patches ranging from 2674.49 to 3261.26 m. The altitude of the lower tree line ranged from 2639.66 to 3220.54 m, and of the upper tree line from 2681.18 to 3412.77 m.

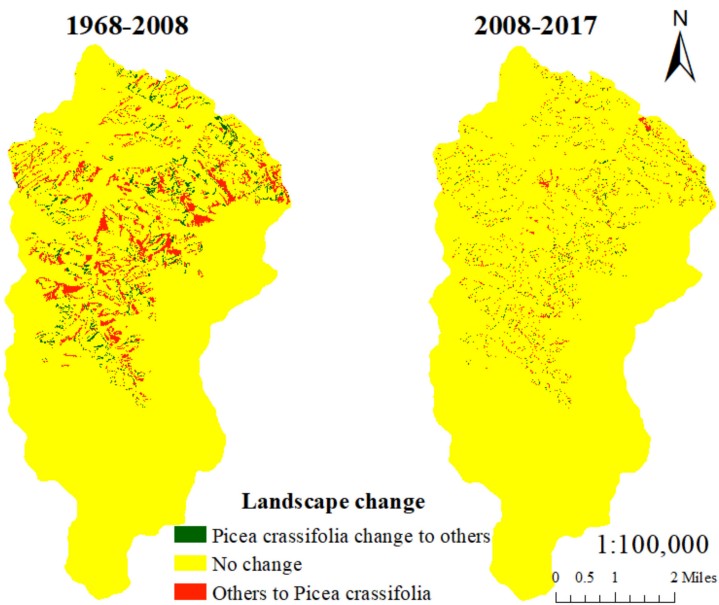

**Figure 5.** Area change of *Picea crassifolia* Kom. from 1968 to 2008 and 2008 to 2017 in the Dayekou watershed.

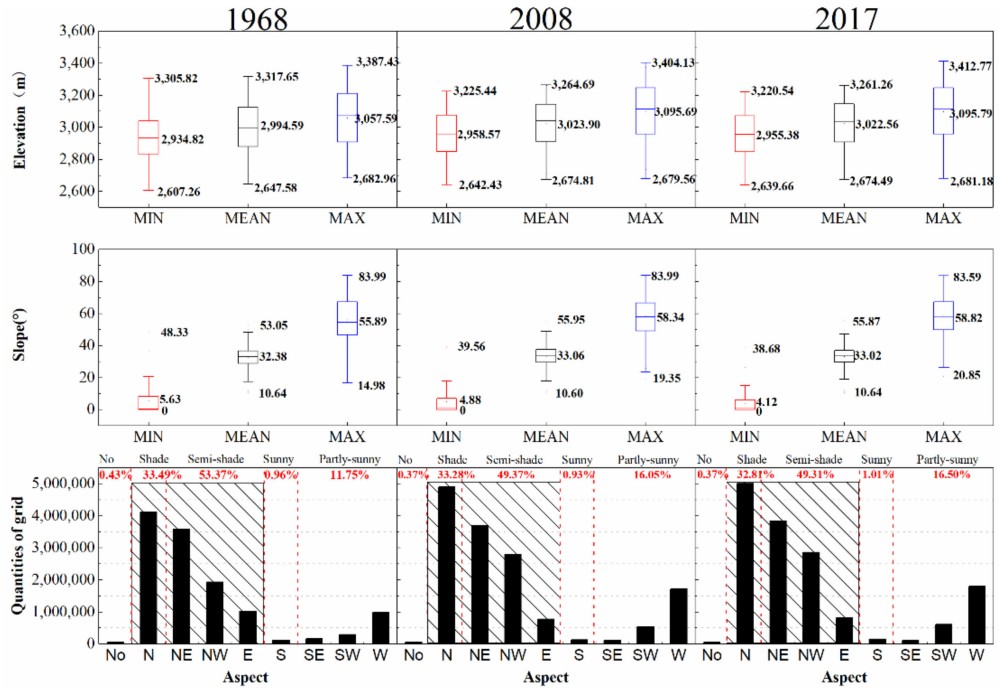

**Figure 6.** Elevation, slope, and aspect features of *Picea crassifolia* Kom. forest distribution.

The average elevation of forest distribution increased from 2994.59 to 3023.90 m from 1968 to 2008, with a rate of increase of 2.44 m/per year, but it declined by 1.34 m and a decline rate of 0.50 m/per year by 2017.

The average altitude of the lower tree line increased by 23.74 m between 1968 and 2008, with a rate of 2.02 m/per year. However, this rate decreased by 1.20 m/per year during 2008–2017.

Throughout the study period, the upper tree line moved upward. During the period 1968–2008, the tree line moved 39.10 m with a rate of 3.12 m/per year. The average rate of the upward movement from 2008–2017 was 0.04 m/per year.

The forest was distributed over a wide range of slopes from 0 to 83.99°. The average slope of forest distribution was about 33°, and slope of the upper tree line, lower tree line, and mean forest did not change significantly. The *Picea crassifolia* Kom. forest was mainly distributed on shaded and semi-shaded slopes, and those accounted for more than 80% of the forest area, reaching 86.86% in 1968. From 1968 to 2017, the proportion of forests distributed on shaded slopes decreased, and the proportion of forests distributed on partly sunny slopes increased significantly.

The elevation distribution index over the past 50 years (Figure 7A) showed *Picea crassifolia* Kom. forests dominated elevations from 2700.00 to 3250.00 m for 50 years. Forest area increased from 1968 to 2008, and the increase was greatest at 2800.00–2900.00 m.

Changes in slope distribution index indicated that *Picea crassifolia* Kom. forest in the past 50 years dominated 10–35° and around 60° slopes (Figure 7B). The forest area on relatively flat slopes increased rapidly from 1968 to 2008. Further, *Picea crassifolia* Kom. forest was mainly found on shaded and semi-shaded slopes; however, the proportion of forest area on shaded slopes decreased between 1968 and 2008, while that on semi-sunny slopes increased (Figure 7C).

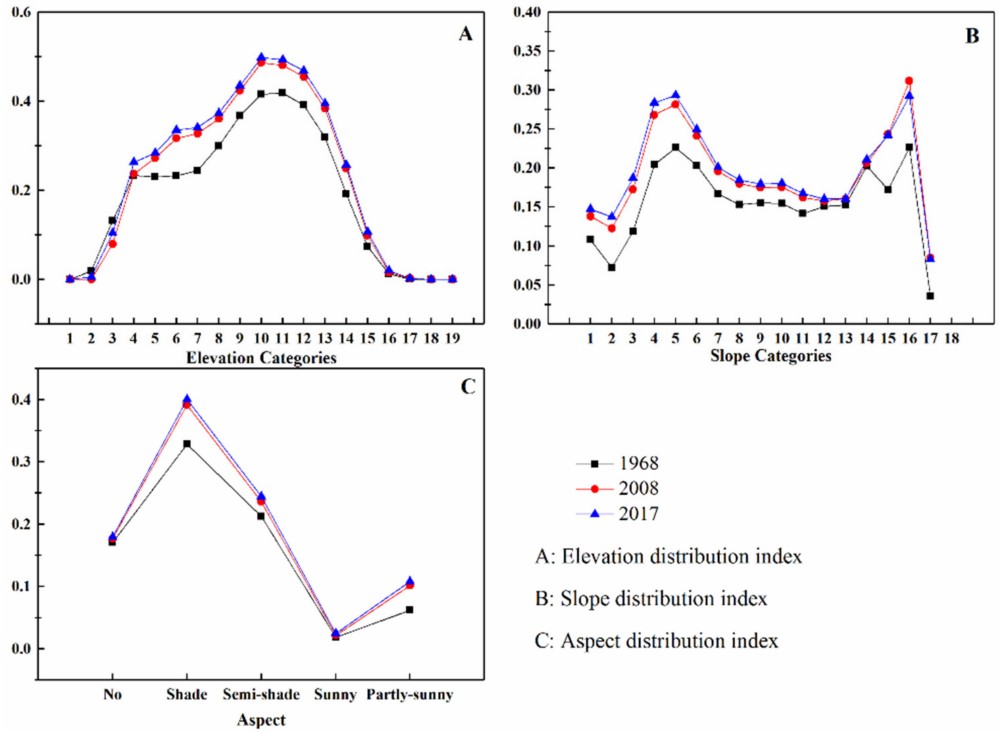

**Figure 7.** Change in the (**A**) elevation, (**B**) slope, and (**C**) aspect distribution index (*y*-axis is the value calculated with Equation (1), and *x*-axis is the levels of elevation, slope, and aspect).

### 3.2. Landscape Changes

The largest patch index (LPI) did not exhibit a clear response function with the change in scale, and there was a large fluctuation at 2, 4, and 5 m (Figure 8). However, beyond 5 m, the exponential change was steady. PAFRAC showed an increasing trend with an increase in scale, and its landscape index scale maps fluctuated slightly at 2, 8, and 10 m. AI showed a downward trend. PD increased between 1 and 6 m and then decreased, with inflection points at 10 and 18 m. Although SPLIT decreased with increasing scale, no apparent relationship was detected, and the trend was opposite of that of LPI. Large fluctuations with a downward trend were found at 2, 4, and 5 m in the SPLIT map, and after 5 m, the exponential change stabilized. AI and LSI decreased linearly with inflection points at 3 and 10 m for AI, and at 4 and 12 m for LSI. Based on the calculated scale effects of the landscape index, the first scale domain of each landscape index and the appropriate scale for research were determined (Table 2). Finally, we found that the most appropriate landscape scale for this study was 7 m.

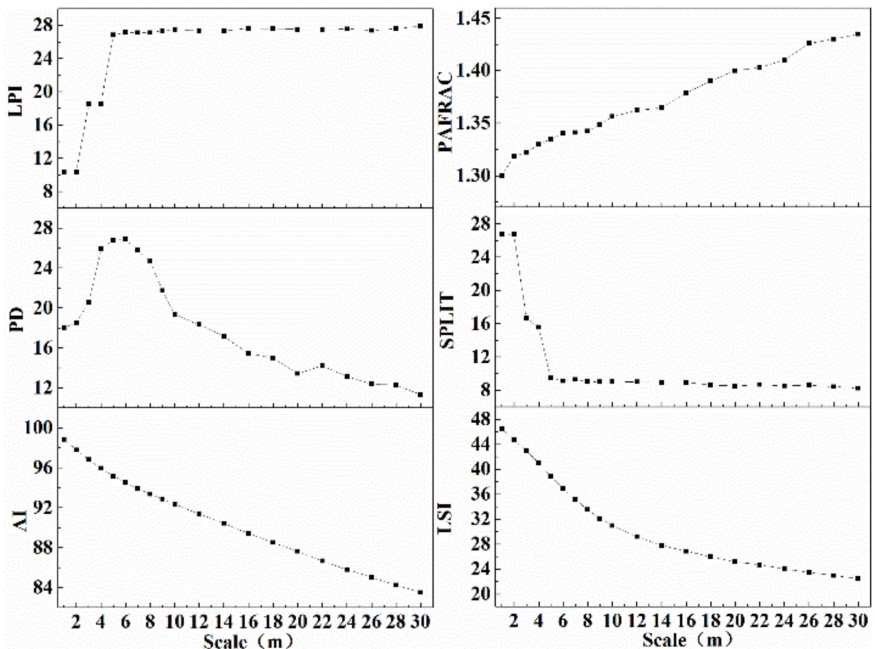

**Figure 8.** Response curve of landscape indices.

**Table 2.** The appropriate scale of the landscape metrics.

| Metrics | First Scale Domain | The Appropriate Scale |
|---------|--------------------|-----------------------|
| LPI | 2–5 m | >5 m |
| PAFRAC | 2–8 m | 3–7 m |
| PD | 6–10 m | 7–9 m |
| SPLIT | 2–5 m | >5 m |
| AI | 3–10 m | 2–9 m |
| LSI | 4–12 m | 3–11 m |
| All | 6–8 m | 7 m |

We used that scale to calculate changes in the six landscape pattern indices over the past 50 years (Figure 9).

The LPI were <30 in the past 50 years, and <20 in 1968, indicating that scattered rather than concentrated patches dominated in the landscape. After 1968, the index increased, but the value remained low, indicating that the development of *Picea crassifolia* Kom. was mainly due to the development of small patches. The calculated value of PAFRAC at about 1.32–1.34 in the past 50 years showed that the shape of forest patches was relatively regular, while the shape of the boundary tended to change from regular to irregular. In the past 50 years, forest PD, that is, patch uniformity, has been decreasing, but the change was relatively small. During last 50 years, SPLIT for the forest decreased, especially from 1968 to 2008, forest patches have developed rapidly, and patches became closer and some even merged. Aggregation Index (AI) did not change greatly in the past 50 years, with a difference from 1968 to 2017 of less than 0.1, indicating that the degree of aggregation did not change. From LSI, patch concentration in the *Picea crassifolia Kom.* forest has increased in the past 50 years, and the distance between patches became smaller, indicating that patches have expanded to a certain extent.

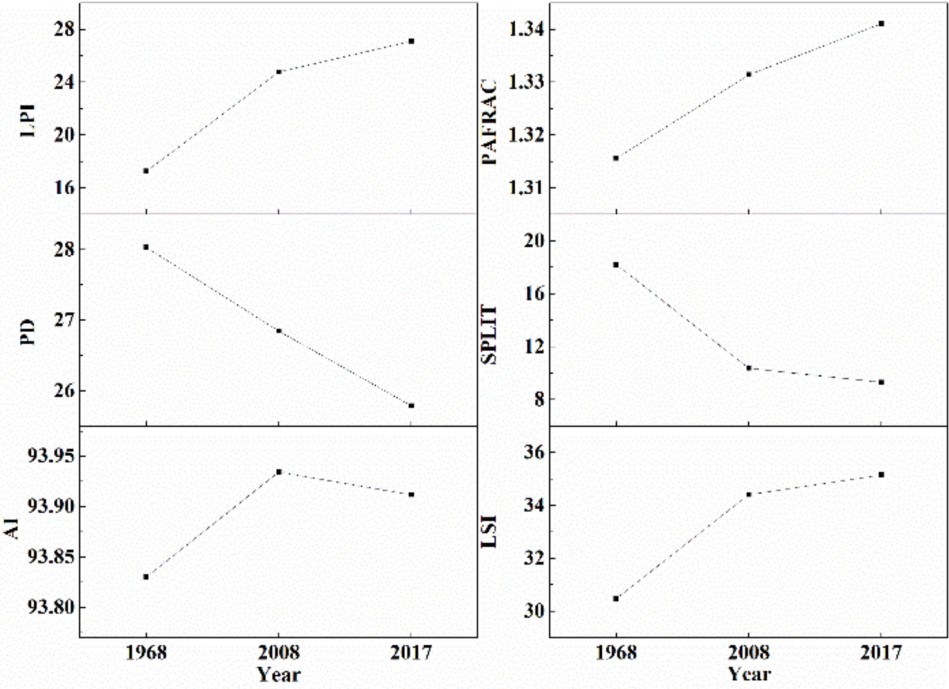

**Figure 9.** The change of landscape pattern.

## 4. Discussion

We used high-resolution remote sensing images for three periods in this study to interpret the distribution of *Picea crassifolia* Kom. forest. Then, we discerned changes in the position of the tree line and attributed them to elevation, slope, aspect, and related indices. We determined characteristics and dynamic changes of the landscape pattern of *Picea crassifolia* Kom. forest with the landscape pattern index. Further, we also analyzed possible reasons for *Picea crassifolia* Kom. forest changes.

### 4.1. Landscape Pattern Characteristics of Picea crassifolia Kom. Forest

We found that *Picea crassifolia* Kom. forest in the Dayekou watershed was mostly distributed in patches and strips. Patchy forests become typical landscape types in arid and semi-arid areas as a result of interactions among accumulation and redistribution of water, nutrients, and other resources, and specific micro-topography [51,52]. High-resolution remote sensing images can monitor this type of vegetation better than other means [53]. The area and aggregation degree of *Picea crassifolia* Kom. forest patches were small. Small patch habitats had more conservation importance than large and medium forest patches [54], but small patches with short distances among them are susceptible to natural disasters and the ecosystem is relatively fragile [55].

The distribution regulation of the spruce forest in Qinghai is more obvious in elevation and aspect than slope. The range of altitudinal distribution of *Picea crassifolia* Kom. forest in Dayekou watershed of 2640 to 3413 m was consistent with earlier reports of the *Picea crassifolia* Kom. range of 2600 to 3400 m in the Qilian Mountains [56]. Our study also showed that *Picea crassifolia* Kom. forest in this watershed dominated at elevations of 2700 to 3250 m for 50 years, which is somewhat higher than the 2650–3100 m observed for the Qilian Mountains [56]. An analysis of stomatal density and its distribution pattern on epidermis, of the length and dry weight of needles, and of tree-ring, showed that altitude of about 3000 m was the optimum zone for growth and development of *Picea crassifolia* Kom. [57]. The apparent preference of *Picea crassifolia* Kom. forests for growth in this watershed on shaded slopes that are either relatively flat and shaded and half-shaded slopes is consistent with forest landscape patterns observed in the Qilian Mountains [58] and are apparently strongly related to humidity and solar radiation [30].

*4.2. Landscape Pattern Change in Dayekou Catchment*

More than 90% of the landscape area in the watershed did not change. However, the number of patches and patch area of the *Picea crassifolia* Kom. forest in Dayekou have been increasing since 1968, especially on flat slopes and partly sunny slope aspects. Rising tree lines have been observed in Europe, North America, and New Zealand [59,60], and at high latitudes and altitudes, 52% of mountain tree lines have shifted upward [61]. The expansion of mountain forests to higher altitudes and latitudes, and the upward movement of vegetation, are also frequently observed in high mountains [62,63]. In our study, forest interior was also vegetated during the study period, especially between 1968 and 2008.

The apparent expansion and upward movement of *Picea crassifolia* Kom. forest in Dayekou differed from the *Picea crassifolia* Kom. forest in Tianshan, northwestern China, where forest change was mainly reflected in an accelerated growth rate, while the tree line was not affected [64]. In Qilian Mountains, NDVI of vegetation appeared to have been increasing during 1982–2014 [65] and the carbon mass of *Picea crassifolia* Kom. forest in the Qilian Mountains has increased by 1.202 kg/m$^2$ from 1964 to 2013 [66]. One survey showed that *Picea crassifolia* Kom. forest population density increased by 23 times at the tree line in the Qilian Mountains, but the tree line position was not significantly altered during the past 100 years [67]. Another survey determined that the elevation of the upper tree line shifted upward between 6.1 to 10.4 m from 1957 to 1980 [16]. We observed in this study an increase in forest density, an expansion of forest area, and an upward trend in the tree line of *Picea crassifolia* Kom. forest in the Qilian Mountains. Research on the Qinghai spruce forests in the Qilian Mountains focus mainly on establishing the relationships between tree growth and climate change through tree-rings [68–70]. Articles discussing changes in *Picea crassifolia* Kom. forest area and landscape pattern are relatively few.

*4.3. Possible Reasons of Picea crassifolia Kom. Forest to Changes*

Historically, forests in the Qilian Mountains underwent deforestation in the 1960s and 1970s [71]. But then, total area of *Picea crassifolia* Kom. forest in Dayekou increased, reflecting the trend of increasing vegetation cover in the Qilian Mountains after 1982 [65], especially following the establishment of the Qilian Mountain Reserve and declaration of the *Picea crassifolia* Kom. forest as an important ecosystem for water conservation and management [72]. Since the 1970s, in some areas of Qilian Mountains, grasslands on semi-shaded and partly sunny slopes had been converted into *Picea crassifolia* Kom. plantation forests [73]. Thus, the growth in *Picea crassifolia* Kom. forest patches and on flat sunny slopes during the study period may be caused by plantation establishment.

In the past 50 years, the average annual temperature and annual precipitation increased in *Picea crassifolia* Kom. growing area in the Qilian Mountains [66]. The increase in temperature due to climate change is more significant in mountainous areas, resulting in expansion of forests [74]. *Picea crassifolia* Kom. growth at the upper tree line in the Qilian Mountains appears to have responded to warmer conditions, too. The slower rate of forest expansion or even a shrinkage after 2008 may be related to the slowing of the temperature rise after 2000 [75,76]. The elevation distribution of the upper tree line of *Picea crassifolia* Kom. forest is mainly related to temperature, while the distribution of the lower tree line is related to precipitation. The rising lower tree line may be due to an increase in evapotranspiration and drought events caused by the temperature increase offsetting the expansion of the lower tree line due to precipitation [77].

## 5. Conclusions

We applied remote sensing images with high resolution to detect the slowly changing *Picea crassifolia* Kom. tree line. We found that the scale of detected change in *Picea crassifolia* Kom. forest was about or slightly below 30 m, which is difficult to monitor with low-resolution remote sensing images, such as Landsat. Therefore, to detect slowly evolving changes in coniferous forests in high mountain areas, it is necessary to use high-resolution imagery.

The number and area of forest patches have been increasing from 1968 to 2017, especially at relatively flat and partly sunny areas. Also, forest patch concentration increased. The patches have expanded, and their upper and lower tree line and average altitude have moved upward. However, the rate of area increase after 2008, and the upward trend of forest lines, started to slow down. Recent changes in *Picea crassifolia Kom.* forests may be related to the establishment of forest plantations and recent climate change.

Because the selected remote sensing image had only three time periods, additional remote sensing data will be added to improve the correlation analysis between forest change and climate factors at a future time.

**Author Contributions:** Z.H. and S.F. conceived and designed the experiments; S.F. performed the experiments, analyzed the data and wrote the paper. All authors have read and agreed to the published version of the manuscript.

**Funding:** This research was funded by the PhD early development program of Shangluo University (19SKY027), the National Key Research and Development Program of China (No.2017YFC0504306), the Strategic Priority Research Program of the Chinese Academy of Sciences (No.Y92C782001), and the National Natural Science Foundation of China (No. 41901050).

**Acknowledgments:** We are very grateful to Kathryn Piatek for her comments and editorial assistance.

**Conflicts of Interest:** The authors declare no conflict of interest. The funding sponsors had no role in the design of the study, data collection, analyses, interpretation, writing of the manuscript, or in the decision to publish the results.

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
