# Peer review of "Fifty Years of Change in a Coniferous Forest in the Qilian Mountains, China—Advantages of High-Definition Remote Sensing"

_forests, doi:10.3390/f11111188_

Round 1

Reviewer 1 Report

This paper analysis the spruce forest limits dynamics in Qilian Mountains based on analysis of high resolution remote sensing image. 

The main concern are related with the climate - landscape change relationship which is not support by any statistical analysis, been very brief presented, but with important conclusion derived. 

Also the conclusions related to spruce plantation impact are not supported by any data. 

More comments are in the attached file. 

Author Response

Point-by-point modification

Dear Review:

Thank you for your comments on our manuscript titled “Fifty years of change in a coniferous forest in the Qilian Mountains, China -- advantages of high-definition remote sensing" (forests-976767). The comments were invaluable for improving our manuscript, and contributed important insights for our further research.

We have carefully addressed the comments, and made corrections to the best of our abilities. We hope that our corrections meet with your full approval. Revised portions of text are marked in “track changes” function (MS Word) in the manuscript. We adjusted the figures. The revised paper has been thoroughly checked by a native speaker for language and consistency. These changes will not influence the content and framework of the paper.

Once again, thank you very much for your comments and suggestions. The main corrections in the paper and responses to academic editor's comments are as follows:

Comments 1, Line19: No analysis were presented on the results part related with the influence of temperature. Remove from abstract; Line25: No clear evidence or statistical analysis on effect of climate change. Remove; Line132: No statistical analysis based on climate data in the paper. remove this part; Line140: Can you detail the climatic variable used? And What type of spatial analysis and which regional statistics were computed? Maybe you can move this sentence at the end of the subchapter; Line 179-181: If you want to keep this part on climate - landscape change relationship more details are needed. For the analysis was used a mean time series for the entire area combining the 12 stations or each station was used for a specific region from the study area? More clarification are needed on this part; Line 279: This part has very low analysis. I suggest to remove the part related with the climate from the paper.  Line 283: decade. Line 345: What is the statistical support of this conclusion? No statistical analysis were conducted to support this conclusion.

Response: Thank you for these helpful suggestions.

Statistical analysis requires long-term continuous remote sensing data with climate data. We only had three periods of high-resolution remote sensing images and have not found a good way to analyze the relationship between forest changes and meteorological factors. We used our best informed judgment about the causes of climate change based on trends in climate and changes in remote sensing data, and did not conduct statistical analysis. Our conclusion of this part in the text is not rigorous.

Thus, based on both reviewers advice, we moved the section about weather factor changes to the discussion for considering meteorological factors as a possible factor that may cause tree line changes.

4.3 Possible reasons of Picea crassifolia Kom. forest to changes

Historically, forests in the Qilian Mountains underwent deforestation in the 1960s and 1970s [1]. But then, total area of Picea crassifolia Kom. forest in Dayekou increased, reflecting the trend of increasing vegetation cover in the Qilian Mountains after 1982 [2], especially following the establishment of the Qilian Mountain Reserve and declaration of the Picea crassifolia Kom. forest as an important ecosystem for water conservation and management [3]. Since the 1970s, in some areas of Qilian Mountains, grasslands on semi-shaded and partly-sunny slopes had been converted into Picea crassifolia Kom. plantation forests[4]. Thus, the growth in Picea crassifolia Kom. forest patches and on flat sunny slopes during the study period may be caused by plantation establishment.

In the past 50 years, the average annual temperature and annual precipitation increased in Picea crassifolia Kom. growing area in the Qilian Mountains [5]. The increase in temperature due to climate change is more significant in mountainous areas, resulting in expansion of forests [6]. Picea crassifolia Kom. growth at the upper tree line in the Qilian Mountains appears to have responded to warmer conditions, too. The slower rate of forest expansion or even a shrinkage after 2008 may be related to the slowing of the temperature rise after 2000 [7,8]. The elevation distribution of the upper tree line of Picea crassifolia Kom. forest is mainly related to temperature, while the distribution of the lower tree line is related to precipitation. The rising lower tree line may be due to an increase in evapotranspiration and drought events caused by the temperature increase offsetting the expansion of the lower tree line due to precipitation [9].

Comments 2, Line21: No quantitative information or distinction in the results part on this topics - plantation.

Response: Thank you for this comment.

Picea crassifolia Kom. is known to grow on shady or semi-shaded slopes; our research showed that it grows mainly in relatively flat and partly-sunny areas. So we concluded that the establishment of plantation forests may be one of the reasons for the changes.

We did not analyze this in a quantitative manner, so we revised the abstract to the following:

The establishment of plantation forests may be one of the reasons for the changes.

Comments 3, Line 46-55:Quite large discussion on spruce dendroclimatology. The aim of the paper are related to remote sensing and tree line advance and not to dendroclimatology. Please consider to shorten this part.

Response: Thank you for pointing this out. We deleted the detailed description on spruce dendroclimatology.

Comments 4, Line 47: Better to use tree growth inside of vegetation growth.

Response: Thank you for pointing this out. We replaced.

Comments 5, Line 89: of ; Line 92: is ; Line114 maybe: 2007? Line 224: not altitude, but slope!!! ; Line 229: ???? ; Line 348: ????

Response: Thank you for pointing this out; we made those changes.

Comments 6, Line 95: this is a type of soil? Please check.

Response: Thank you for pointing this out. We corrected the type of soil.

Comments 7, Line 206, 207,218: maybe per year; per year. Unclear? take in consideration to change in text with year.

Response: Thank you for pointing this out. We Changed m/a to m/per year.

Comments 8, Line357-358: Not supported by the results and statistical analysis; Line 360: This conclusion isn't supported by data. No information are presented related with the amount of spruce plantation in the area; Line 360: Which correlation? In the results part are not presented correlation with climate variable.

Response: Thank you for pointing this out. Based on this, and the other reviewer’s advice, we rewrote the conclusion section as follows:

We applied remote sensing images with high resolution to detect the slowly-changing Picea crassifolia Kom. tree line. We found that the scale of detected change in Picea crassifolia Kom. forest was about or slightly below 30 m, which is difficult to monitor with low-resolution remote sensing images, such as Landsat. Therefore, to detect slowly-evolving changes in coniferous forests in high mountain areas, it is necessary to use high-resolution imagery.

The number, and area of forest patches have been increasing from 1968 to 2017, especially at relatively flat and partly sunny areas. Also, forest patch concentration increased. The patches have expanded, and their upper and lower tree line and average altitude have moved upward. However, the rate of area increase after 2008, and the upward trend of forest lines started to slow down. Recent changes in Picea crassifolia Kom. forests may be related to the establishment of forest plantations and recent climate change.

Because the selected remote sensing image had only three time periods, additional remote sensing data will be added to improve the correlation analysis between forest change and climate factors at a future time.

  1. Liu, Z.; Zhao, C.; Bai, Y.; Peng, S.; Nan, Z.; Liu, X.; Hao, H. Difference in stem volume of Qinghai spruce (Picea crassifolia) in catchments of Qilian Mountains [in Chinese with English abstract]. Journal of Lanzhou University (Natural Sciences) 2013, 747-751.
  2. Jia, W.; Chen, J. Variations of NDVI and Its Response to Climate Change in the Growing Season of Vegetation in Qilianshan Mountains from 1982 to 2014 [in Chinese with English abstract]. Research of Soil and Water Conservation 2018, 25, 264-268.
  3. Zhao, C.; Bie, Q.; Peng, H. Analysis of the Niche Space of Picea crassifolia on the Northern Slope of Qilian Mountains [in Chinese with English abstract]. Acta Geographica Sinica 2010, 113-121.
  4. Zhu, X.; He, Z.; Chen, L.; Du, J.; Yang, J.; Lin, P.; Li, J. Changes in Species Diversity, Aboveground Biomass, and Distribution Characteristics along an Afforestation Successional Gradient in Semiarid Picea crassifolia Plantations of Northwestern China. Forest Science 2016.
  5. Fang, S.; He, Z.; Du, J.; Chen, L.; Lin, P.; Zhao, M. Carbon Mass Change and Its Drivers in a Boreal Coniferous Forest in the Qilian Mountains, China from 1964 to 2013. Forests 2018, 9, 57.
  6. de Wit, H.A.; Bryn, A.; Hofgaard, A.; Karstensen, J.; Peters, G.P. Climate warming feedback from mountain birch forest expansion: reduced albedo dominates carbon uptake. Global Change Biology 2014, 20, 2344.
  7. Wang, X.; Chen, R.; Liu, J. Spatial and Temporal Variation Characteristics of Accumulated Negative Temperature in Qilian Mountains under Climate Change [in Chinese with English abstract]. Plateau Meteorology 2017, 36, 1267-1275.
  8. Cao, G.; Fu, J.; Li, L.; Cao, S.; Tang, Z.; Jiang, G.; Yu, M.; Yuan, J.; Han, G.; Diao, E. Analysis on Temporal and Spatial Variation Characteristics of Air Temperature in the South Slope of Qilian Mountains and Its Nearby Regions During the Period From 1960 to 2014 [in Chinese with English abstract]. Research of Soil and Water Conservation 2018, 25, 88-96.
  9. Yang, W.; Y, W.; AA, W.; Z, L.; X, T.; Z, H.; S, W.; P, Y. Influence of climatic and geographic factors on the spatial distribution of Qinghai spruce forests in the dryland Qilian Mountains of Northwest China. Science of the Total Environment 2018, 612, 1007.

Reviewer 2 Report

The authors used historical remote sensing images to detect high altitude forest change in the Qilian Mountains, China. Similar studies are critical to understanding forest response to climate change.

Major Comments

Authors have reported that tree line and forest line change in the study, but they did not define what tree line and forest line is. And how these variables were measured in the remote sensing images used in the study.

Objective 3 “forest change relationship with climate change” needs significant revision. How relationship analysis was done is not presented in the MS. I suggest the authors present detailed analysis as supplementary information. Temperature and precipitation are not the only variables controlling the tree line position. There are multiple factors controlling the tree line position (see Korner 2012, Holtmeier 2005), so it will vague to report that temperature change is slowly responsible for tree line advance.

The discussion section needed to be rewritten again. At the current stage, it is missing discussion related to tree line change, forest change, and relation with climatic factors.

There are numerous formatting issues in the MS and it is making MS difficult to follow. MS needs to be checked thoroughly for addressing the formatting issues.

Provide more descriptive figures captions – provide the full form of abbreviations, use a scale bar for all of your figures.

Minor Comments

 Line 49. Use tree-ring instead of tree ring

Line 65. Tree-line or tree line be consistent

Line 68, 73. LANDSAT or Landsat?

Line 71, 74, 76, and rest of the MS. Make species name italic - Picea crassifolia

Line 84. the

Line 86. km2

Line 89-90. Fix the formatting issue

Line 88-91. Are these climate data is from the nearby meteorological station? How far from the study forest, and at what elevation?

Figure 1. what is the yellow color in the map representing? I suggest using a bar scale or both (bar scale and ration scale)

Line 101-102. The authors are using three different spatial resolutions 1.8 m, 2.4 m, 2.1 m remote sensing images in the study. Did the author use the resolution matching technique to match the resolution of remote sensing images from three sources?

Line 102. Please elaborate high data quality

Line 116. What are terrain factors, and how you extracted them?

Figure 2. Use Elevation (m) instead of DEM in legend

Line 123. How topographic influence (shadow etc.) were removed from these three images?

Line 131. Space between 2017were

Figure 4. Use Elevation (m) in place of DEM in legend. Use bar sale or both bar scale and ratio scale

Line 140. ESRI ArcMap 10.4

Line 141. Provide detail about topographic data – how you generate them?

Line 149, 150. 19, eighteen be consistent

Line 149, 150. Describe why 19 and 18 classes

Line 160. It is not clear what is topographic factors of elevation, slope, and aspect. Please provide detail

Line 180. How these data were analyzed? 12 stations average or separately? Provide detail as supplementary information

Line 181. Provide the detail of how the relationship between climate and forest position was analyzed? Correlation and regression analysis or other?

Line 184. Shift in the tree line or Shift in the forest cover?

Figure 5. One of the objectives of this study was to detect tree line change. I suggest the authors map or show tree line in these three images

Line 206 and 207. m2

Line 213. law of elevation?

Line 214-216, 235-236, and rest of the MS. Decimal place after elevation (2639.70 m) or 2700? be consistent

Line 216. Lower tree line, upper tree line. Please provide how you will define the tree line? And how you detected tree line in three images you used in this study

Line 218, 220. m /a or m/a?

Line 238. 60-80° slope is too steep for the forest to establish

Figure 8. Use elevation and slope in figure a and b X-axis

Line 252. 5 m

Line 280-283. Are these temperatures and precipitation increased rate for one station or an average of 12 stations?

Line 290-293. This sentence is too long. Please break down into smaller sentences

Line 304. distribution low?

Line 317. Please rewrite the sentence – did not no change

Line 335. Tree-ring

Line 345. It will be very vague to make a statement that temperature is the main factor responsible for forest change. There are multiple factors controlling forest growth. The current study does not account lack of disturbance, soil. ecology, and physiology etc.

Line 348. 200lizes?

Line 352. I suggest using detect instead of monitor

Line 353-354. I see a contrasting statement in this sentence. If the accurate scale is 30 m, most of the recent Landsat images (spatial resolution is 28.5 – 30 m) will fall under this category. However, in many places in this MS authors mention Landsat is low resolution and we can use for forest change detection

Line 353. Forest line or tree line?

Author Response

Dear Review:

Thank you for your comments on our manuscript titled “Fifty years of change in a coniferous forest in the Qilian Mountains, China -- advantages of high-definition remote sensing" (forests-976767). The comments were invaluable for improving our manuscript, and contributed important insights for our further research.

We have carefully addressed the comments, and made corrections to the best of our abilities. We hope that our corrections meet with your full approval. Revised portions of text are marked in “track changes” function (MS Word) in the manuscript. We adjusted the figures. The revised paper has been thoroughly checked by a native speaker for language and consistency. These changes will not influence the content and framework of the paper.

Once again, thank you very much for your comments and suggestions. The main corrections in the paper and responses to academic editor's comments are as follows:

Comment1:Authors have reported that tree line and forest line change in the study, but they did not define what tree line and forest line is. And how these variables were measured in the remote sensing images used in the study.

Response: Thank you for pointing this out.  We added the definition of the tree line and the measurement of remote sensing images in the introduction as follows:

The tree line is the continuous forest boundary of a mountain forest, and it is the transition zone from forest to tundra [1-3].

As early as 1995, tree lines of patch forest ecotones in the Rocky Mountain National Park were detected with aerial photographs of 1988 and 1990 [4]. The tree line can be directly detected based on the vegetation type over the remote sensing image, and the identification of the tree line can be more accurate with the use of the Normalized Difference Vegetation Index (NDVI) or topographic data such as elevation and slope [5,6]. Direct classification for Landsat images can be used for extracting tree line ecotone, but a complex method, such as linear spectral mixture analysis, should be used for detecting the alpine tree lines [7,8].

Comment 2: Objective 3 “forest change relationship with climate change” needs significant revision. How relationship analysis was done is not presented in the MS. I suggest the authors present detailed analysis as supplementary information. Temperature and precipitation are not the only variables controlling the tree line position. There are multiple factors controlling the tree line position (see Korner 2012, Holtmeier 2005), so it will vague to report that temperature change is slowly responsible for tree line advance; minor comments: Line 88-91: Are these climate data is from the nearby meteorological station? How far from the study forest, and at what elevation?; Line 180: How these data were analyzed? 12 stations average or separately? Provide detail as supplementary information; Line 181: Provide the detail of how the relationship between climate and forest position was analyzed? Correlation and regression analysis or other?; Line 280-283: Are these temperatures and precipitation increased rate for one station or an average of 12 stations?; Line 345. It will be very vague to make a statement that temperature is the main factor responsible for forest change. There are multiple factors controlling forest growth. The current study does not account lack of disturbance, soil. ecology, and physiology etc.

Response: Thank you for these helpful suggestions.

Statistical analysis requires long-term continuous remote sensing data with climate data. We only had three periods of high-resolution remote sensing images and have not found a good way to analyze the relationship between forest changes and meteorological factors. We used our best informed judgment about the causes of climate change based on trends in climate and changes in remote sensing data, and did not conduct statistical analysis. Our conclusion of this part in the text is not rigorous.

Thus, based on both reviewers advice, we moved the section about weather factor changes to the discussion for considering meteorological factors as a possible factor that may cause tree line changes.

4.3 Possible reasons of Picea crassifolia Kom. forest to changes

Historically, forests in the Qilian Mountains underwent deforestation in the 1960s and 1970s [9]. But then, total area of Picea crassifolia Kom. forest in Dayekou increased, reflecting the trend of increasing vegetation cover in the Qilian Mountains after 1982 [10], especially following the establishment of the Qilian Mountain Reserve and declaration of the Picea crassifolia Kom. forest as an important ecosystem for water conservation and management [11]. Since the 1970s, in some areas of Qilian Mountains, grasslands on semi-shaded and partly-sunny slopes had been converted into Picea crassifolia Kom. plantation forests[12]. Thus, the growth in Picea crassifolia Kom. forest patches and on flat sunny slopes during the study period may be caused by plantation establishment.

In the past 50 years, the average annual temperature and annual precipitation increased in Picea crassifolia Kom. growing area in the Qilian Mountains [13]. The increase in temperature due to climate change is more significant in mountainous areas, resulting in expansion of forests [14]. Picea crassifolia Kom. growth at the upper tree line in the Qilian Mountains appears to have responded to warmer conditions, too. The slower rate of forest expansion or even a shrinkage after 2008 may be related to the slowing of the temperature rise after 2000 [15,16]. The elevation distribution of the upper tree line of Picea crassifolia Kom. forest is mainly related to temperature, while the distribution of the lower tree line is related to precipitation. The rising lower tree line may be due to an increase in evapotranspiration and drought events caused by the temperature increase offsetting the expansion of the lower tree line due to precipitation [17].

Comment 3: The discussion section needed to be rewritten again. At the current stage, it is missing discussion related to tree line change, forest change, and relation with climatic factors.

Response: Thank you for pointing this out. We revised the discussion section, and added relevant content in sections 4.2 and 4.3.

  • Landscape pattern change in Dayekou catchment

More than 90% of the landscape area in the watershed did not change. However, the number of patches and patch area of the Picea crassifolia Kom. forest in Dayekou have been increasing since 1968, especially on flat slopes and partly-sunny slope aspects. Rising tree lines have been observed in Europe, North America, and New Zealand [18,19]; at high latitudes and altitudes, 52% of mountain tree lines have shifted upward [20]. The expansion of mountain forests to higher altitudes and latitudes, and the upward movement of vegetation are also frequently observed in high mountains [21,22]. In our study, forest interior was also vegetated during the study period, especially between 1968 and 2008.

The apparent expansion and upward movement of Picea crassifolia Kom. forest in Dayekou differed from the Picea crassifolia Kom. forest in Tianshan, northwestern China, where forest change was mainly reflected in an accelerated growth rate, while the tree line was not affected [23]. In Qilian Mountains, NDVI of vegetation appeared to have been increasing during 1982-2014 [10] and the carbon mass of Picea crassifolia Kom. forest in the Qilian Mountains has increased by 1.202 kg/m2 from 1964 to 2013 [13]. One survey showed that Picea crassifolia Kom. forest population density increased by 23 times at the tree line in the Qilian Mountains, but the tree line position was not significantly altered during the past 100 years [24]. Another survey determined that the elevation of the upper tree line shifted upward between 6.1 to 10.4 m from 1957 to 1980 [25]. We observed in this study an increase in forest density, an expansion of forest area, and an upward trend in the tree line of Picea crassifolia Kom. forest in the Qilian Mountains. Research on the Qinghai spruce forests in the Qilian Mountains focus mainly on establishing the relationships between tree growth and climate change through tree-rings [26-28]; articles discussing changes in Picea crassifolia Kom. forest area and landscape pattern are relatively few.

Comment 4: There are numerous formatting issues in the MS and it is making MS difficult to follow. MS needs to be checked thoroughly for addressing the formatting issues.

Response: Thank you for pointing this out. We revised the manuscript carefully to conform to the formatting.

Comment 5: Provide more descriptive figures captions – provide the full form of abbreviations, use a scale bar for all of your figures.

Response: Thank you for pointing this out. Based on this, and the other comments, we modified all figures and used bar scale and ration scale both for all of my figures.

Comments 6, Line 49: Use tree-ring instead of tree ring Line 335: Tree-ring

Response: Thank you for this comment. Based on the second reviewer’s comment, we actually deleted the tree-ring research description, and we replaced tree ring with tree-ring in other parts.

Comments 7, Line 65: Tree-line or tree line be consistent; Line 353:  Forest line or tree line?

Response: Thank you for pointing this out. We replaced the ‘tree-line’, ‘forest line’ with tree-line in the entire text.

Comments 8, Line 68, 73: LANDSAT or Landsat?

Response: Thank you for these helpful suggestions. We used Landsat in full text.

Comments 9, Line 71, 74, 76, and rest of the MS: Make species name italic - Picea crassifolia

Response: Thank you for pointing this out. We checked the rest of the MS and made species name italic.

Comments 10, Line 84: the; Line 86: km2 ;Line 89-90: Fix the formatting issue; Line 131: Space between 2017were; Line 140: Line 149, 150. 19, eighteen be consistent; Line 206 and 207: m2; Line 252: 5 m; Line 348. 200lizes?

Response: Thank you for these helpful suggestions. We made these changes.

Comments 11, Figure 1: what is the yellow color in the map representing? I suggest using a bar scale or both (bar scale and ration scale); Figure 2: Use Elevation (m) instead of DEM in legend; Figure 4: Use Elevation (m) in place of DEM in legend. Use bar sale or both bar scale and ratio scale; Figure 5: One of the objectives of this study was to detect tree line change. I suggest the authors map or show tree line in these three images; Figure 8: Use elevation and slope in figure a and b X-axis

Response: Thank you for these helpful suggestions.  We clarified information in the figures. We deleted Figure 4 and showed tree line in figure 5.

Comments 12, Line 101-102: The authors are using three different spatial resolutions 1.8 m, 2.4 m, 2.1 m remote sensing images in the study. Did the author use the resolution matching technique to match the resolution of remote sensing images from three sources?

Response: Thank you for this comment. Remote sensing resolution image matching is generally applied to images with relatively large differences [29]. We did not match the resolution for we think that the resolution difference between the three remote sensing images was very small.

Comments 13, Line 102. Please elaborate high data quality, Line 123: How topographic influence (shadow etc.) were removed from these three images?

Response: Our remote sensing images are customized by a company that specializes in processing remote sensing images, and we have selected the images with the best remote sensing image quality (the least cloud amounts among satellite images of similar resolution for the same period.) in the same period. At the same time, due to technical limitations, especially for historical aerial photographs, we have ordered remote sensing image preprocessing services. Therefore, the software we used for analysis already carried out preprocessing such as atmospheric correction, geometric correction, data fusion, and greenness adjustment.

Comments 14, Line 116: What are terrain factors, and how you extracted them? ; Line 141: Provide detail about topographic data – how you generate them? Line 149, 150: Describe why 19 and 18 classes; Line 160: It is not clear what is topographic factors of elevation, slope, and aspect. Please provide detail

Response: Thank you for these helpful suggestions.

We chose elevation, slope, and aspect as the main terrain factors.

We used the dem data and calculated slope and aspect by ‘Surface’ in ‘Spatial Analyst Tools’ in ESRI ArcMap 10.4. We used ‘Reclassify’ in ‘Spatial Analyst Tools’ to classify of elevation, slope, aspect. Based on our selected classification scale and data range, elevation and slope data are divided into 18 and 19 categories.

To clarify this in text, we modified the corresponding part 2.3.1

We used three terrain indicators - elevation, slope, and aspect which were calculated by ‘Surface’ tool in ‘Spatial Analyst Tools’ in ESRI ArcMap 10.4 - to detect dynamic changes in forest distribution. Then we used ‘Reclassify’ in ‘Spatial Analyst Tools’ to classify elevation, slope, aspect.

Comments 15, Line 140: ESRI ArcMap 10.4

Response: We corrected it.

Comments 16, Line 184: Shift in the tree line or Shift in the forest cover?

Response: We clarified this.

Comments 17, Line 213: law of elevation? Line 304: distribution low?

Response: Thank you for point this out. We changed low to regulation.

Comments 18, Line 214-216, 235-236: and rest of the MS. Decimal place after elevation (2639.70 m) or 2700? be consistent

Response: Thank you for pointing this out. We unified the format of the numbers in the article.

Comments 19, Line 216: Lower tree line, upper tree line. Please provide how you will define the tree line? And how you detected tree line in three images you used in this study

Response: Thank you for these helpful suggestions. We have added relevant definitions and methods in the Materials and Methods section 2.2 as follows:

The boundary of the forest is the tree line. We have calculated the maximum, minimum, and average values of each forest patch by ‘Zonal statistics’ tool in ‘Spatial Analyst Tools’ in ESRI ArcMap 10.4 to explore the upper, lower, and overall tree line changes.

Comments 20, Line 218, 220: m /a or m/a?

Response: Thank you for pointing this out. Based on your and other reviewer’ advice, we changed all unit to m/per year.

Comments 21, Line 238: 60-80° slope is too steep for the forest to establish

Response: We consulted the literature, but did not find information on slope distribution of  Picea crassifolia Kom. forest in the Qilian Mountains. Related research is generally small in scale, and addressed  classification of altitude and aspect. During our field investigation, we found that the upper tree line of Picea crassifolia Kom. forest does exist on a relatively steep slope (the following figure).

We revised the relevant parts of the article based on your opinion.

Examples of  upper tree lines

Comments 22, Line 290-293: This sentence is too long. Please break down into smaller sentences

Response: We agree, and we addressed this.

We used high-resolution remote sensing images for three periods in this study to interpret the distribution of Picea crassifolia Kom. forest. Then we discerned changes in the position of the tree line and attributed them to elevation, slope, aspect, and related indices. Then, we determined characteristics and dynamic changes of the landscape pattern of Picea crassifolia Kom. forest using the landscape pattern index. Further, we also analyzed possible reasons for Picea crassifolia Kom. forest changes.

Comments 23, Line 317: Please rewrite the sentence – did not no change

Response: Thank you for pointing this out. We deleted the ‘no’.

More than 90% of the landscape area in the watershed did not change

Comments 24, Line 352: I suggest using detect instead of monitor

Response: We made that change.

Comments 25, Line 353-354: I see a contrasting statement in this sentence. If the accurate scale is 30 m, most of the recent Landsat images (spatial resolution is 28.5 – 30 m) will fall under this category. However, in many places in this MS authors mention Landsat is low resolution and we can use for forest change detection

Response: Thank you for pointing this out. 

Landsat images are often used to detect changes in vegetation, but it’s hard to detect alpine tree line which grow slowly and continuously due to Landsat classification issues with outliers and temporal inconsistency[7,8]. The forest we chose has changed very slowly at a rate of 30 m(less than one pixel of Landsat image) in 50 years. If we use Landsat images, it is likely that such small difference over 50 years will not be detected. Thus, we chose data with a resolution of about 2m. For forests with relative tree species, high-resolution images should be considered when selecting data, otherwise vegetation change may be missed.

In this part of the article, we made the following amendments to clarify the conclusions:

Based on this, and the other reviewer’s advice, we rewrote the conclusion section as follows:

We applied remote sensing images with high resolution to detect the slowly-changing Picea crassifolia Kom. tree line. We found that the scale of detected change in Picea crassifolia Kom. forest was about or slightly below 30 m, which is difficult to monitor with low-resolution remote sensing images, such as Landsat. Therefore, to detect slowly-evolving changes in coniferous forests in high mountain areas, it is necessary to use high-resolution imagery.

The number, and area of forest patches have been increasing from 1968 to 2017, especially at relatively flat and partly sunny areas. Also, forest patch concentration increased. The patches have expanded, and their upper and lower tree line and average altitude have moved upward. However, the rate of area increase after 2008, and the upward trend of forest lines started to slow down. Recent changes in Picea crassifolia Kom. forests may be related to the establishment of forest plantations and recent climate change.

Because the selected remote sensing image had only three time periods, additional remote sensing data will be added to improve the correlation analysis between forest change and climate factors at a future time.

  1. Körner, C. A re-assessment of high elevation treeline positions and their explanation. Oecologia 1998, 115, 445-459.
  2. Wieser, G. Lessons from the timberline ecotone in the Central Tyrolean Alps: a review. Plant Ecology & Diversity 2012, 5, 127-139.
  3. Hofgaard, A.; Dalen, L.; Hytteborn, H. Tree recruitment above the treeline and potential for climate‐driven treeline change. Journal of Vegetation Science 2009, 20, 1133-1144.
  4. Baker, W.L.; Honaker, J.J.; Weisberg, P.J. Using aerial photography and GIS to map the forest-tundra ecotone in Rocky Mountain National Park, Colorado, for global change research. Photogrammetric Engineering and Remote Sensing 1995, 61, 313-320.
  5. Panigrahy, S.; Anitha, D.; Kimothi, M.; Singh, S. Timberline change detection using topographic map and satellite imagery. Tropical Ecology 2010, 51, 87-91.
  6. Olthof, I.; Pouliot, D. Treeline vegetation composition and change in Canada's western Subarctic from AVHRR and canopy reflectance modeling. Remote Sensing of Environment 2010, 114, 805-815.
  7. Xu, D.; Geng, Q.; Jin, C.; Xu, Z.; Xu, X. Tree Line Identification and Dynamics under Climate Change in Wuyishan National Park Based on Landsat Images. Remote Sensing 2020, 12, 2890.
  8. Chen, Y.; Lu, D.; Luo, G.; Huang, J. Detection of vegetation abundance change in the alpine tree line using multitemporal Landsat Thematic Mapper imagery. International Journal of Remote Sensing 2015, 36, 4683-4701.
  9. Liu, Z.; Zhao, C.; Bai, Y.; Peng, S.; Nan, Z.; Liu, X.; Hao, H. Difference in stem volume of Qinghai spruce (Picea crassifolia) in catchments of Qilian Mountains [in Chinese with English abstract]. Journal of Lanzhou University (Natural Sciences) 2013, 747-751.
  10. Jia, W.; Chen, J. Variations of NDVI and Its Response to Climate Change in the Growing Season of Vegetation in Qilianshan Mountains from 1982 to 2014 [in Chinese with English abstract]. Research of Soil and Water Conservation 2018, 25, 264-268.
  11. Zhao, C.; Bie, Q.; Peng, H. Analysis of the Niche Space of Picea crassifolia on the Northern Slope of Qilian Mountains [in Chinese with English abstract]. Acta Geographica Sinica 2010, 113-121.
  12. Zhu, X.; He, Z.; Chen, L.; Du, J.; Yang, J.; Lin, P.; Li, J. Changes in Species Diversity, Aboveground Biomass, and Distribution Characteristics along an Afforestation Successional Gradient in Semiarid Picea crassifolia Plantations of Northwestern China. Forest Science 2016.
  13. Fang, S.; He, Z.; Du, J.; Chen, L.; Lin, P.; Zhao, M. Carbon Mass Change and Its Drivers in a Boreal Coniferous Forest in the Qilian Mountains, China from 1964 to 2013. Forests 2018, 9, 57.
  14. de Wit, H.A.; Bryn, A.; Hofgaard, A.; Karstensen, J.; Peters, G.P. Climate warming feedback from mountain birch forest expansion: reduced albedo dominates carbon uptake. Global Change Biology 2014, 20, 2344.
  15. Wang, X.; Chen, R.; Liu, J. Spatial and Temporal Variation Characteristics of Accumulated Negative Temperature in Qilian Mountains under Climate Change [in Chinese with English abstract]. Plateau Meteorology 2017, 36, 1267-1275.
  16. Cao, G.; Fu, J.; Li, L.; Cao, S.; Tang, Z.; Jiang, G.; Yu, M.; Yuan, J.; Han, G.; Diao, E. Analysis on Temporal and Spatial Variation Characteristics of Air Temperature in the South Slope of Qilian Mountains and Its Nearby Regions During the Period From 1960 to 2014 [in Chinese with English abstract]. Research of Soil and Water Conservation 2018, 25, 88-96.
  17. Yang, W.; Y, W.; AA, W.; Z, L.; X, T.; Z, H.; S, W.; P, Y. Influence of climatic and geographic factors on the spatial distribution of Qinghai spruce forests in the dryland Qilian Mountains of Northwest China. Science of the Total Environment 2018, 612, 1007.
  18. Parmesan, C.; Yohe, G. A globally coherent fingerprint of climate change impacts across natural systems. Nature 2003, 421, 37-42.
  19. Pauli, H.; Gottfried, M.; Reiter, K.; Klettner, C.; Grabherr, G. Signals of range expansions and contractions of vascular plants in the high Alps: observations (1994-2004) at the GLORIA* master site Schrankogel, Tyrol, Austria. Global Change Biology 2007, 13, 147-156.
  20. Harsch, M.A.; Hulme, P.E.; Mcglone, M.S.; Duncan, R.P. Are treelines advancing? A global meta-analysis of treeline response to climate warming. Ecology Letters 2009, 12, 1040.
  21. MacDonald, G.; Kremenetski, K.; Beilman, D. Climate change and the northern Russian treeline zone. Philosophical Transactions of the Royal Society of London B: Biological Sciences 2008, 363, 2283-2299.
  22. Wilson, R.J.; Gutiérrez, D.; Gutiérrez, J.; Martínez, D.; Agudo, R.; Monserrat, V.J. Changes to the elevational limits and extent of species ranges associated with climate change. Ecology Letters 2005, 8, 1138-1146.
  23. Qi, Z.; Liu, H.; Wu, X.; Hao, Q. Climate‐driven speedup of alpine treeline forest growth in the Tianshan Mountains, Northwestern China. Global change biology 2015, 21, 816-826.
  24. Zhang, L.; Liu, H. Response of Picea crassifolia Population to Climate Change at the Treeline Ecotones in Qilian Mountains[in Chinese with English abstract]. SCIENTIA SILVAE SINICAE 2012, 18-21.
  25. He, Z.; Zhao, W.; Zhang, L.; Liu, H. Response of tree recruitment to climatic variability in the alpine treeline ecotone of the Qilian Mountains, northwestern China. Forest Science 2013, 59, 118-126.
  26. Wang, B.; Chen, T.; Li, C.; Xu, G.; Wu, G.; Liu, G. Radial growth of Qinghai spruce (Picea crassifolia Kom.) and its leading influencing climate factor varied along a moisture gradient. Forest Ecology and Management 2020, 476, 118474.
  27. Liang, E.; Leuschner, C.; Dulamsuren, C.; Wagner, B.; Hauck, M. Global warming-related tree growth decline and mortality on the north-eastern Tibetan plateau. Climatic Change 2016, 134.
  28. Gou, X.; Chen, F.; Yang, M.; Li, J.; Peng, J.; Jin, L. Climatic response of thick leaf spruce (Picea crassifolia) tree-ring width at different elevations over Qilian Mountains, northwestern China. Journal of Arid Environments 2005, 61, 513-524.
  29. Dufournaud, Y.; Schmid, C.; Horaud, R. Matching images with different resolutions. In Proceedings of Proceedings IEEE Conference on Computer Vision and Pattern Recognition. CVPR 2000 (Cat. No. PR00662); pp. 612-618.

Round 2

Reviewer 1 Report

The authors responded to all my review suggestion.